# Detection and Evaluation of Blast Resistance Genes in Backbone *Indica* Rice Varieties from South China

**DOI:** 10.3390/plants13152134

**Published:** 2024-08-01

**Authors:** Liqun Tang, Jian Song, Yongtao Cui, Honghuan Fan, Jianjun Wang

**Affiliations:** Institute of Crops and Nuclear Technology Utilization, Zhejiang Academy of Agricultural Sciences, Hangzhou 310021, China; tanglq@zaas.ac.cn (L.T.); songj@zaas.ac.cn (J.S.); cuiyt@zaas.ac.cn (Y.C.); xixi615@163.com (H.F.)

**Keywords:** disease resistance, molecular markers, *indica* rice, rice blast management, resistance gene pyramiding

## Abstract

Rice blast caused by the pathogenic fungus *Magnaporthe oryzae* poses a significant threat to rice cultivation. The identification of robust resistance germplasm is crucial for breeding resistant varieties. In this study, we employed functional molecular markers for 10 rice blast resistance genes, namely *Pi1*, *Pi2*, *Pi5*, *Pi9*, *Pia*, *Pid2*, *Pid3*, *Pigm*, *Pikh*, and *Pita*, to assess blast resistance across 91 *indica* rice backbone varieties in South China. The results showed a spectrum of resistance levels ranging from highly resistant (HR) to highly susceptible (HS), with corresponding frequencies of 0, 19, 40, 27, 5, and 0, respectively. Yearly correlations in blast resistance genes among the 91 key *indica* rice progenitors revealed *Pid2* (60.44%), *Pia* (50.55%), *Pita* (45.05%), *Pi2* (32.97%), *Pikh* (4.4%), *Pigm* (2.2%), *Pi9* (2.2%), and *Pi1* (1.1%). Significant variations were observed in the distribution frequencies of these 10 resistance genes among these progenitors across different provinces. Furthermore, as the number of aggregated resistance genes increased, parental resistance levels correspondingly improved, though the efficacy of different gene combinations varied significantly. This study provides the initial steps toward strategically distributing varieties of resistant *indica* rice genotypes across South China.

## 1. Introduction

Rice (*Oryza sativa* L.) production confronts significant threats from diseases and pests, with rice blast, commonly known as “rice cancer”, emerging as a major obstacle. The hemibiotrophic fungal pathogen *Magnaporthe oryzae* triggers rice blast, which is among the most widespread and damaging diseases impacting rice cultivation [1,2,3,4]. Given its extensive geographical spread and adaptability to diverse environmental conditions, yield losses attributed to this fungal infection range between 10% and 30% [2,5]. This results in the annual devastation of significant rice harvests sufficient to sustain over 60 million individuals and incurs economic losses exceeding $70 billion [6]. Recognizing its significant threat to both agricultural productivity and scientific advancement, *Science Magazine* classified rice blast in 2010 as one of the “most severe bioterrorism threats to food safety” [1]. Moreover, the international molecular plant pathology community ranked the rice blast fungus at the forefront of the “top 10 plant pathogenic fungi”, underscoring its significance [7,8]. In recent years, the incidence rate of rice blast in China has gradually increased, causing increasingly severe damage to rice. Almost all rice-growing areas in China have experienced rice blast, especially in the southern rice-growing areas with a hot and humid climate, which is more conducive to the spread of rice blast [9].

Integrating resistance (R) genes into high-quality rice varieties to develop resistant cultivars has proven to be the most environmentally friendly and sustainable strategy for managing rice blast [4]. Over the past decades, researchers have identified over 350 loci associated with quantitative resistance (QRLs) and 100 rice blast-related resistance genes, many of which have been cloned and functionally characterized [5,10,11], such as *Pi1*, *Pi2*, *Pi5*, *Pi9*, *Piz-t*, *Pigm*, *Pikm*, *Pikh*, *Pish*, *Pia*, *Pita*, *Pi21*, *Pit*, *Pi25*, *Pi35*, *Pi36*, *Pi37*, *Pi50*, *Pi56*, *Pi64*, *Pi-d2*, *Pi-d3*, and so on [12,13,14]. The majority of blast resistance genes in rice exhibit dominance and qualitative resistance, distributed on all 12 chromosomes [15]. Using loss and gain of function approaches, researchers have confirmed that *Pi1*, a broad-spectrum resistance gene against *Magnaporthe oryzae*, is an allele at the *Pik* locus [16]. Marker-assisted selection (MAS) was employed to introduce *Pi1*, *Pi2*, and *D12* into Jin23B and its hybrid rice varieties, Jinyou 402 and Jinyou 207. The results showed that the more resistance genes contained in Jin23B and its hybrid rice, the stronger the resistance to rice blast [17]. Using a sequence-based allele mining approach, researchers amplified and sequenced the crucial resistant genes *Pi2*/*Pi9*, present as allelic variants on chromosome 6, across 361 disease-resistant varieties [18]. Previous studies have established that *PID2*, which encodes a B-lectin receptor-like kinase, plays a crucial role in conferring resistance to rice against the strain ZB15 of *Magnaporthe oryzae* [19]. Thus far, only a limited number of R genes have been recognized as conferring broad-spectrum resistance, such as *Piz*, *Piz-t*, *Pi1*, *Pi2*, *Pi9*, *Pi33*, *Pi54*, *Pigm*, and *Pi40*. One of these genes, *Pigm*, was identified from the Chinese local variety Gumei 4, known for its strong and lasting resistance to rice blast. *Pigm* showed resistance against 50 different isolates of *M. oryzae* originating from various regions worldwide [13,20,21,22].

The widespread application of chemical pesticides to combat rice blast not only entails substantial expenses but also results in environmental contamination and excessive pesticide residues, posing risks to ecological balance and food safety. Consequently, there is widespread recognition that consolidating multiple blast resistance genes and cultivating highly resistant varieties represents the most cost-effective, eco-friendly, and efficient strategy for preventing and controlling rice blast disease. By employing molecular marker-assisted selection, conventional hybridization, and high-intensity stress screening, three broad-spectrum blast resistance genes, *Pi9*, *Pi5*, and *Pi54*, were incorporated into the blast-resistant variety Huhan 1S. This resulted in the creation of a novel water-saving and drought-resistant PTGMS line known as Huhan 74S, which exhibited a resistance frequency of 85.7% against 14 prevalent blast isolates [23]. Near-isogenic and backcross lines of Koshihikari rice cultivars containing single or two-gene combinations of blast resistance genes (*Pi21*, *Pi34*, and *Pi35*) were utilized to assess their efficacy in managing leaf blast. The results demonstrated that the combination of two resistance genes showed superior effectiveness in reducing leaf blast compared to a single gene [24]. Pyramiding of *Pi54*+*Pi1*, *Piz5*+*Pi54*, *Pi9*+*Pita*, *Pib*+*Pi25*+*Pi54*, and *Pi9*+*Pi2*+*Piz* and these combinations have shown effective and durable resistance to blast, demonstrating their potential to enhance rice resistance against *Magnaporthe oryzae* pathogen [25,26,27,28,29]. These research findings provide a sound theoretical reference for the selection and rational layout of resistant varieties. In this study, functional molecular markers targeting 10 blast resistance genes, namely *Pi1*, *Pi2*, *Pi5*, *Pi9*, *Pia*, *Pid2*, *Pid3*, *Pigm*, *Pikh*, and *Pita*, were utilized to identify and assess the presence of resistance genes within 91 foundational varieties in *indica* rice cultivated in South China. The primary aim was to investigate the occurrence of these genes among the backbone varieties in rice planting areas in South China and their correlation with blast resistance. This analysis sought to establish a theoretical framework and provide valuable parental resources with blast resistance for the strategic integration of resistance genes in breeding programs.

## 2. Results

### 2.1. Variability of Resistance to Rice Blast

Laboratory analyses conducted over two consecutive years (2022–2023) indicated that the comprehensive resistance level distributions of the tested parental lines were highly resistant (HR), resistant (R), moderately resistant (MR), moderately susceptible (MS), susceptible (S), and highly susceptible (HS) at 0, 19, 40, 27, 5, and 0, respectively. Among the 91 varieties, their proportions were 0%, 21.98%, 43.96%, 29.67%, 17.58%, and 0%, respectively. Findings revealed consistent resistance levels among the evaluated parental lines over the years, with 59 varieties (64.84%) exhibiting resistance to seedling blight. Across the same evaluation period, resistance levels among the backbone parental lines showed minimal variation, indicating high repeatability of results (Table 1).

### 2.2. Identification and Analysis of Rice Blast Resistance Genes among the Backbone Rice Varieties in South China

This study detected the distribution of 10 rice blast resistance genes, including *Pi1*, *Pi2*, *Pi5*, *Pi9*, *Pia*, *Pid2*, *Pid3*, *Pigm*, *Pikh*, and *Pita*, among the backbone varieties of *indica* rice in South China (Table 1 and Figure 1). The identification results revealed varied distribution frequencies of the 10 resistance genes among 91 varieties. *Pid3* exhibited the highest distribution frequency at 70.33%, followed by *Pi5* at 62.64%. Distribution frequencies of *Pid2*, *Pia*, and *Pita* were 60.44%, 50.55%, and 45.05%, respectively. For the remaining genes, distribution frequencies decreased successively: 32.97% for *Pi2*, 4.4% for *Pikh*, 2.2% for *Pigm*, 2.2% for *Pi9*, and 1.1% for *Pi1* (Figure 1).

In terms of the number of resistance genes carried, all the varieties of the backbone *indica* rice carried one or more resistance genes (Table 1). Among them, varieties harboring between two and four resistance genes were 17, 32, and 21, respectively, accounting for 76.92% of the total. Five varieties carried a single resistance gene, accounting for 5.49%. There were 12 varieties with five resistance genes, accounting for 13.19%, and a minimum of 4 varieties carried six resistance genes, accounting for 4.4% (Figure 2a). Further analysis revealed that all six varieties from Sichuan carried a number of resistance genes between two and five, constituting 100% of the cohort, with no detection of varieties carrying only a single or as many as six genes. In Hunan, of the seven cultivated materials, two varieties carried three resistance genes, one carried four, and two carried five, accounting for 28.57%, 14.29%, and 28.57%, respectively. In Zhejiang Province, among 18 varieties, 17 carried between two and five resistance genes, with distribution percentages of 22.22%, 33.33%, 22.22%, and 16.67%, while one variety had a single gene, accounting for 5.56%. In Guangdong Province, all 51 varieties possessed between one and six resistance genes, with 18 varieties carrying three genes, representing the largest subgroup at 35.29%. The number of varieties carrying four resistance genes was the second subgroup, with a total of 15 varieties, accounting for 29.41%. The number of varieties carrying two, five, six, and one resistance genes decreased successively, and the number of backbone varieties was seven, six, three, and two, accounting for 13.73%, 11.76%, 5.88%, and 3.92%, respectively (Figure 2b).

In regional distribution, *Pia*, *Pid2*, and *Pid3* genes were widely distributed in Sichuan, Hunan, Zhejiang, and Guangdong. The distribution frequency of *Pi5* was high in Hunan, Zhejiang, and Guangdong, but it was not detected in Sichuan Province. The distribution frequency of the 10 resistance genes was different in different regions. For example, in *indica* rice varieties grown in Sichuan, the distribution frequencies of *Pia*, *Pid2*, *Pid3*, *Pigm*, *Pikh*, and *Pita* genes were 83.33%, 50%, 83.33%, 33.33%, 16.67%, and 16.67%, respectively. The distribution frequencies of *Pi5*, *Pi9*, *Pia*, *Pid2*, *Pid3*, and *Pita* genes were 71.43%, 28.57%, 42.86%, 57.14%, 57.14%, and 28.57%, respectively. The distribution frequencies of *Pi2*, *Pi5*, *Pia*, *Pid2*, *Pid3*, and *Pita* genes were 22.22%, 72.22%, 61.11%, 61.11%, 66.67%, and 38.89%, respectively. In Guangdong, the distribution frequencies of *Pi2*, *Pi5*, *Pia*, *Pid2*, *Pid3*, and *Pita* genes were higher, which were 41.18%, 70.59%, 45.10%, 60.78%, 70.59%, and 56.86%, respectively. Furthermore, the *Pi1* gene was not detected in all varieties from Sichuan, Hunan, Zhejiang, and Guangdong, while the *Pi9* gene was exclusively identified in varieties from Hunan. The *Pigm* gene was solely detected in varieties from Sichuan (Figure 3). 

### 2.3. Correlation Analysis of Gene Combination Distributions and Resistance Phenotypes

To explore the correlation between the types or number of resistance genes in the tested varieties and their resistance phenotypes to rice blast, we conducted an analysis on the distribution of 10 blast resistance genes in the 91 selected core varieties and their resistance phenotypes to seedling blast. The results showed that among the core varieties carrying one to six rice blast resistance gene frequencies, 40%, 58.82%, 62.50%, 66.68%, 75%, and 100%, respectively, exhibited disease resistance. With an increase in the number of aggregated resistance genes, the resistance level of the varieties showed an increase accordingly. According to the evaluation of seedling blast resistance over two years, most of the varieties carrying between one and three resistance genes showed a moderate resistance or susceptibility level with a general resistance performance. There were a total of 59 varieties with a seedling blast resistance level of moderate resistance or higher, including 4 from Sichuan, 4 from Hunan, 14 from Zhejiang, 32 from Guangdong, and Hubei, Jiangxi, Anhui, Fujian, and Guangxi all had only 1 variety (Table 2).

Further analysis was conducted to evaluate how the presence of 10 distinct resistance genes, *Pi1*, *Pi2*, *Pi5*, *Pi9*, *Pia*, *Pid2*, *Pid3*, *Pigm*, *Pikh*, and *Pita*, affected the frequency of resistance within varieties. The results revealed significant variances in the contribution of each resistance gene to enhancing blast resistance in rice parental lines. Among the variety lines harboring the *Pi1* gene, only one exhibited a blast resistance frequency of 0%, while variety lines carrying *Pi9*, *Pigm*, and *Pikh* genes had frequencies of blast resistance at 100%, 100%, and 75%, respectively, involving two, three, and four lines each. Parental lines carrying *Pi2* (30 varieties), *Pi5* (57 varieties), *Pia* (46 varieties), *Pid2* (55 varieties), *Pid3* (64 varieties), and *Pita* (41 varieties), exhibited blast resistance frequencies of 83.33%, 68.42%, 69.56%, 63.63%, 59.38%, and 73.17%, respectively (Table 2 and Figure 4). Despite the low prevalence of *Pi9* and *Pigm* genes among the tested lines, with frequencies of 2.2% and 3.3%, respectively, these genes showed promising resistance, highlighting their potential utility in rice blast resistance breeding across Hunan, Sichuan, and Anhui. Except for *Pi1*, varieties carrying *Pi2*, *Pi5*, *Pia*, *Pid2*, *Pid3*, *Pikh*, and *Pita* demonstrated strong blast resistance performance, suggesting potential utility in rice blast resistance breeding in Southern China. In summary, among the 10 different resistance genes *Pi1*, *Pi2*, *Pi5*, *Pi9*, *Pia*, *Pid2*, *Pid3*, *Pigm*, *Pikh*, and *Pita*, their contributions to resistance breeding were ranked as follows: *Pigm*, *Pi9*, *Pi2*, *Pikh*, *Pita*, *Pia*, *Pi5*, *Pid2*, and *Pid3*.

Accounting for the resistance exhibited by rice varieties against rice blast, an analysis was conducted to evaluate how different combinations of resistance genes contribute to enhancing rice’s resistance. Varieties with identical sets of resistance genes were categorized together, resulting in a total of 32 gene combinations. The combination of *Pi5*+*Pia*+*Pid2*+*Pid3*+*Pita* was the most prevalent among rice varieties (nine varieties, accounting for 9.9%), exhibiting a rice blast resistance rate of 77.8%, followed closely by the rice varieties carrying the gene combination of *Pia*+*Pid2*+*Pid3* (seven varieties, accounting for 7.7%), with a rice resistance rate of 42.9%. Rice varieties possessing 20 distinct combinations of resistance genes exhibited moderate to high resistance levels against blast. Among them, the gene combinations of *Pi2*+*Pi5*+*Pia*+*Pid2*+*Pid3*+*Pita* (four varieties, rice blast resistance rate of 100%), *Pi2*+*Pi5*+*Pia* (three varieties, rice blast resistance rate of 100%), *Pi2*+*Pita* (three varieties, rice blast resistance rate of 100%), *Pi2*+*Pid2*+*Pid3* (two varieties, rice blast resistance rate of 100%), and *Pi2*+*Pi5*+*Pid2*+*Pid3* (two varieties, rice blast resistance rate of 100%) emerged as the top five effective combinations of resistance genes (Table 3). This study also found that varieties carrying the same rice blast-resistant gene types exhibited consistent resistance performance, indicating that the resistance gene combination is a key factor determining the level of rice blast resistance in different rice varieties.

## 3. Discussion

### 3.1. Distribution and Manifestation of Blast Resistance Genes in Rice Varieties

Rice blast is a significant threat to rice cultivation, with substantial economic and food security implications. Developing broad-spectrum resistant varieties through gene pyramiding necessitates a comprehensive understanding of the genetic resistance traits in rice varieties. Previous studies have identified functional markers for *Pik*, *Pik*-m, *Pik*-p, *Pi1*, and *Pi64* within China’s core germplasm. *Pik* was found in a single variety (Fidongtang rice), while *Pik-p* was detected in two varieties (Wenxiangnuo) and *Pik-m* in two varieties (Cungunuo and Zhengdao 5) [16,30,31]. The resistance of rice varieties in the Nanumpet and Mandya regions of Karnataka, India, against rice blast was evaluated. It was found that the single gene frequencies of 20 major blast resistance genes ranged from 10.34% to 100.00%, with *Pi9* and *Pizt* genes showing relatively low and high distribution frequencies, respectively [32].

This study focused on broad-spectrum resistance genes including *Pi1*, *Pi2*, *Pi5*, *Pi9*, *Pia*, *Pid2*, *Pid3*, *Pigm*, *Pikh*, and *Pita*. The distribution of these genes was analyzed in 91 *indica* rice backbone varieties using the functional molecular marker technique, and it was found that the *Pid3* gene exhibited the highest distribution frequency, reaching 70.33%, followed by the *Pi5* gene at 62.64%. The distribution frequencies of the *Pid2*, *Pia*, and *Pita* genes were 60.44%, 50.55%, and 45.05%, respectively. *Pid3*, *Pi5*, and *Pid2* genes were widely used in rice breeding in Southern China, while *Pia*, *Pita*, and *Pi2* genes were less distributed. The occurrence rate of *Pikh*, *Pigm*, *Pi9*, and *Pi1* genes was notably low. Specifically, *Pikh* was identified in *indica* rice backbone varieties, just two from Sichuan and two from Guangdong, while the *Pigm* gene was found in two *indica* rice backbone varieties from Sichuan, and the *Pi9* gene was observed in two *indica* rice backbone varieties from Hunan. As for the *Pi1* gene, it was detected only in one parent of the *indica* rice backbone variety originating from Hubei Province. The unbalanced distribution of resistance genes indicated that *Pikh*, *Pigm*, *Pi9*, and *Pi1* were not fully utilized in the conventional breeding process of *indica* rice in South China, and different resistance genes should be combined to improve the resistance of rice varieties [33]. 

### 3.2. Gene Pyramiding for Broad-Spectrum Rice Blast Resistance

Pyramiding or combining multiple resistance genes has proven effective in developing rice varieties with durable resistance to *M. oryzae*. The interaction of various broad-spectrum resistance genes, such as *Pi1*, *Pi2*, *Pi5*, *Pi9*, *Pia*, *Pid2*, *Pid3*, *Pigm*, *Pikh*, and *Pita*, needs further exploration to understand their cooperative mechanisms fully. By evaluating different alleles at the *Piz* locus and their interactions with other R genes, we found that Polygene Pyramiding Lines (PPLs) showed significantly higher resistance levels than monogenic lines [34]. For instance, combining *Pi9* and *Pi54* in near-isogenic lines using molecular marker-assisted selection revealed enhanced blast resistance compared to lines with single genes [35]. Similarly, introducing *Pi46* and *Pita* into the elite restorer line HH179 broadened its resistance spectrum [36]. In our study, functional molecular marker technology was used to identify 10 rice blast resistance genes. Results showed varied effectiveness in rice blast resistance: varieties with *Pi1* showed no resistance, while those with *Pi9*, *Pigm*, and *Pikh* exhibited 100%, 100%, and 75% resistance frequencies, respectively. Varieties with *Pi2* (30 varieties), *Pi5* (57 varieties), *Pia* (46 varieties), *Pid2* (55 varieties), *Pid3* (64 varieties), and *Pita* (41 varieties) had resistance frequencies of 83.33%, 68.42%, 69.56%, 63.63%, 59.38%, and 73.17%, respectively (Table 2). Despite the low prevalence of *Pi9* and *Pigm*, their high resistance performance underscores their potential utility in breeding programs [33,37,38]. 

However, this study also revealed that the number of resistance genes does not necessarily correlate with higher resistance levels. For instance, genotypes 23P-81 and 23P-84, despite having multiple genotypes, did not exhibit superior resistance. Similarly, lines with identical resistance genes, such as 23P-36 and 23P-41, showed different resistance performances. These discrepancies suggest that gene interactions, rather than the number of resistance genes, are crucial for effective resistance. Additionally, some materials, like those carrying *Pigm* and *Pikh* genes, were limited by small sample sizes, making it challenging to evaluate their resistance performance objectively. This limitation indicates the need for further studies with larger sample sizes to accurately assess the effectiveness of these genes. This study demonstrated that combining resistance genes can expand the resistance spectrum and improve physiological resistance. However, it is essential to select genes with complementary resistance spectra for pyramiding to avoid redundant resistance effects. For example, parental materials 23P-4 and 23P-5 from Sichuan, harboring *Pia*, *Pid2*, *Pid3*, *Pigm*, and *Pikh*, exhibited similar resistance levels, while materials 23P-71 and 23P-72 from Guangdong, combining *Pi2*, *Pi5*, *Pia*, *Pid2*, *Pid3*, and *Pita*, showed different results, high lighting the complexity of gene interactions.

Currently, breeding durable and broad-spectrum resistant rice varieties remains a cost-effective strategy for controlling rice blast. This study identified several useful blast-resistant genes, including *Pi1*, *Pi2*, *Pi5*, *Pi9*, *Pia*, *Pid2*, *Pid3*, *Pigm*, *Pikh*, and *Pita*, for further utilization. These genes can be targeted for incorporation into major cultivated rice varieties, providing data support for developing new rice varieties resistant to rice blast. Future breeding efforts should aim to integrate the genetic basis of disease resistance with practical field resistance. Combining marker-assisted selection (MAS) and conventional breeding has enabled the incorporation of these broad-spectrum R genes into elite rice varieties, enhancing their resistance and durability against rice blast. 

## 4. Materials and Methods

### 4.1. Materials and Strains

The 91 high-quality *indica* rice varieties tested were from Sichuan, Hubei, Hunan, Jiangxi, Anhui, Fujian, Zhejiang, Guangdong, Guangxi Province, and other regions. All materials were cultivated at the Institute of Crops and Nuclear Technology Utilization, Zhejiang Academy of Agricultural Sciences, Hangzhou, Zhejiang Province. After evaluation of agronomic traits such as plant type, growth duration, rice quality, yield, and rice blast resistance, these lines were deemed suitable for the Zhejiang rice-growing area and selected as breeding backbone varieties for this study. Detailed parental information is shown in Appendix A.

Nine physiological strains (ZA63, ZB9, ZC15, ZG1, ZB31, EC13, ZB1, EC9, and EZ15) were selected for rice blast resistance detection. All strains were single-spore isolated from Jinggangshan, Daweishan, Taojiang, and Shanghang in China. Each strain was selected based on pathogenicity tests conducted on physiological races of rice blast in the preceding year. All strains were preserved in the Institute of Crop and Nuclear Technology Utilization of Zhejiang Academy of Agricultural Sciences.

### 4.2. Resistance Identification

Laboratory identification of resistance to rice blast in seedlings.

The isolates were isolated from a single spore, and the number of inoculated strains was 9. Previous studies have provided the separation method of blast spores and laboratory identification method [39,40]. After soaking the test material with antibiotic 402 for 1 d, it was rinsed with water to promote germination for 1~2 d, sowed in the seedling tray, and 15~20 seeds were sowed for each variety. When the seedling grew to 3 leaves, the test material was sprayed with rice blast strain spore mixed suspension. Twenty-four hours after inoculation, the arch holder was covered with a plastic film to moisturize and a sunshade net was used to prevent direct sunlight. The incidence was investigated 10 d after inoculation according to the International Rice Research Institute (IRRI), grade 0 is highly resistant, grade 1–2 is resistant, grade 3 is moderately resistant, grade 4–5 is moderately sensitive, grade 6–7 is sensitive, and grade 8–9 is highly sensitive [41].

### 4.3. Rice Blast Resistance Gene Detection

Previous studies have informed the choice of these particular genes due to their known association with rice blast resistance in rice or other relevant crops [18,19,20,23,33,34,42]. This selection strategy ensures a comprehensive assessment of the genetic factors contributing to rice blast resistance, providing valuable insights for breeding programs and genetic improvement efforts. So 10 mainstream rice blast resistance genes, including *Pi1*, *Pi2*, *Pi5*, *Pi9*, *Pia*, *Pid2*, *Pid3*, *Pigm*, *Pikh*, and *Pita*, were chosen in this study. DNA was extracted by the conventional CTAB method, and the PCR reaction system was 10 μL at 35 cycles. Molecular-specific labeling primers were provided by Wuhan Jingtai Biotechnology Co., Ltd., Wuhan, China. After the completion of PCR, FAM and HEX were used as the report fluorescence, ROX was used as the reference fluorescence, and the fluorescence signal was read by TECAN infinite F200 enzymoscope (Tecan, Zürich, Switzerland). The fluorescence signal was analyzed and converted to obtain a clear and intuitive typing map, and genotype results were output according to different colors. The specific test was commissioned by Wuhan Jingtai Biotechnology Co., Ltd., Wuhan, China. Markers utilized in this study are detailed in Appendix A.

## Figures and Tables

**Figure 1 plants-13-02134-f001:**
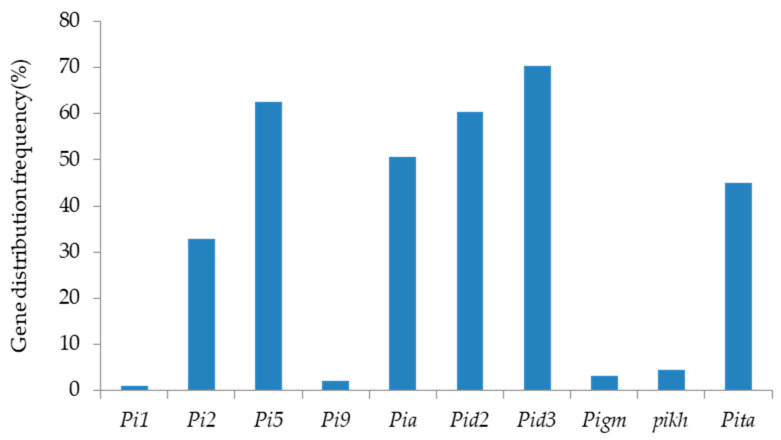
Distribution of blast resistance genes among backbone varieties of *indica* rice.

**Figure 2 plants-13-02134-f002:**
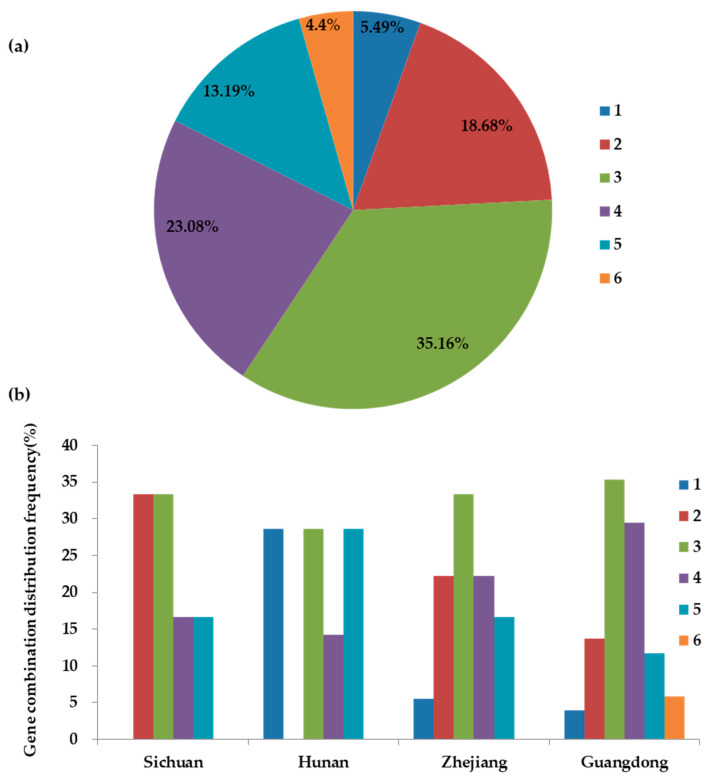
Percentage and distribution of blast resistance gene combinations among backbone varieties of *indica* rice. (**a**) Percentage of various blast resistance gene combinations. (**b**) Spatial distribution frequency of blast resistance combination. Numbers 1–6: The number of blast resistance genes among different regions.

**Figure 3 plants-13-02134-f003:**
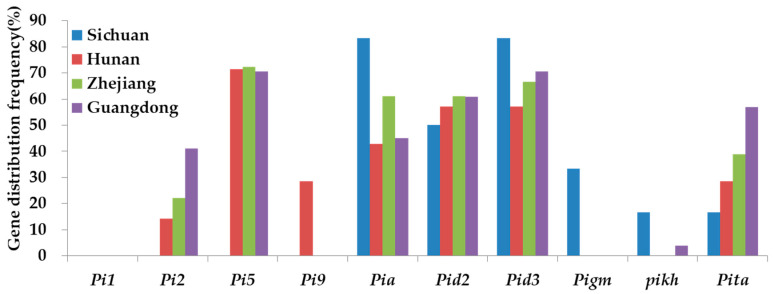
Geographical distribution of rice blast resistance genes across different provinces.

**Figure 4 plants-13-02134-f004:**
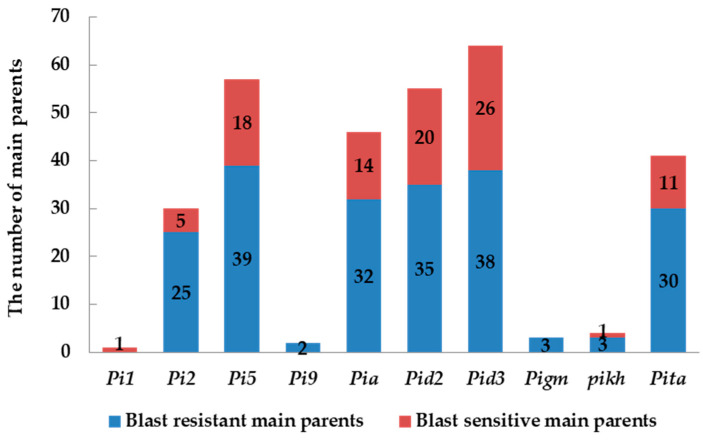
Correlation analysis of gene type with blast resistance in backbone varieties of *indica* rice.

**Table 1 plants-13-02134-t001:** Seedling blast resistance in backbone varieties of *indica* rice.

Number	Germplasm Origin	Rice Blast Resistance Gene	Blast Resistance Level
23P-1	Sichuan	* Pia*+*Pid2*+*Pid3*	MR
23P-2	Sichuan	* Pia*+*Pid3*	MS
23P-3	Sichuan	* Pid3*+*Pita*	MR
23P-4	Sichuan	* Pia*+*Pid2*+*Pid3*+*Pigm*	R
23P-5	Sichuan	* Pia*+*Pid2*+*Pid3*+*Pigm*+*pikh*	R
23P-6	Sichuan	* Pia*+*Pid3*+*pikh*	MS
23P-7	Hubei	* Pi1*+*Pi5*	MS
23P-8	Hubei	* Pi2*+*Pid2*+*Pid3*	MR
23P-9	Hunan	* Pi2*+*Pi5*+*Pita*	MS
23P-10	Hunan	* Pi5*+*Pia*+*Pid2*+*Pid3*+*Pita*	MR
23P-11	Hunan	* Pia*	MS
23P-12	Hunan	* Pi5*+*Pi9*+*Pid2*+*Pid3*	MR
23P-13	Hunan	* Pi5*+*Pi9*+*Pia*+*Pid2*+*Pid3*	R
23P-14	Hunan	* Pi5*	MR
23P-15	Hunan	* Pia*+*Pid2*+*Pid3*	MS
23P-16	Jiangxi	* Pi2*+*Pi5*	MS
23P-17	Jiangxi	* Pid2*+*Pid3*	MS
23P-18	Jiangxi	* Pi2*+*Pi5*+*Pia*+*Pid2*+*Pid3*+*Pita*	MR
23P-19	Anhui	* Pia*+*Pid2*+*Pid3*	MS
23P-20	Anhui	* Pia*+*Pigm*	MR
23P-21	Fujian	* Pi2*+*Pid2*+*Pid3*	R
23P-22	Zhejiang	* Pi5*+*Pia*+*Pid2*+*Pid3*+*Pita*	R
23P-23	Zhejiang	* Pi5*+*Pia*	R
23P-24	Zhejiang	* Pi2*+*Pi5*+*Pita*	R
23P-25	Zhejiang	* Pi2*+*Pi5*	MR
23P-26	Zhejiang	* Pi5*+*Pid2*+*Pid3*+*Pita*	MR
23P-27	Zhejiang	* Pid2*+*Pid3*+*Pita*	MR
23P-28	Zhejiang	* Pia*+*Pid2*+*Pid3*	MS
23P-29	Zhejiang	* Pi5*+*Pid2*+*Pid3*	MS
23P-30	Zhejiang	* Pia*+*Pid2*+*Pid3*	R
23P-31	Zhejiang	* Pi5*+*Pid3*	MS
23P-32	Zhejiang	* Pi5*+*Pia*+*Pid2*+*Pid3*+*Pita*	MR
23P-33	Zhejiang	* Pi5*+*Pia*+*Pid2*+*Pid3*	MR
23P-34	Zhejiang	* Pi2*+*Pi5*	MR
23P-35	Zhejiang	* Pi5*+*Pia*+*Pid2*+*Pid3*+*Pita*	MR
23P-36	Zhejiang	* Pia*+*Pid2*+*Pid3*	MS
23P-37	Zhejiang	* Pia*	MR
23P-38	Zhejiang	* Pi2*+*Pi5*+*Pia*+*Pita*	MR
23P-39	Zhejiang	* Pi5*+*Pia*+*Pid2*+*Pid3*	MR
23P-40	Guangdong	* Pi2*+*Pi5*+*Pita*	R
23P-41	Guangdong	* Pia*+*Pid2*+*Pid3*	MR
23P-42	Guangdong	* Pid3*	MS
23P-43	Guangdong	* Pi5*+*Pia*+*Pid2*+*Pid3*+*Pita*	MR
23P-44	Guangdong	* Pi5*+*Pid2*+*Pid3*	MS
23P-45	Guangdong	* Pi2*+*Pi5*+*Pia*+*Pid2*+*Pid3*+*Pita*	MR
23P-46	Guangdong	* Pi2*+*Pi5*+*Pia*	R
23P-47	Guangdong	* Pid2*+*Pid3*+*Pita*	MS
23P-48	Guangdong	* Pid2*+*Pid3*+*Pita*	MS
23P-49	Guangdong	* Pi2*+*Pi5*+*Pia*	MR
23P-50	Guangdong	* Pi5*+*Pid2*+*Pid3*	MR
23P-51	Guangdong	* Pi5*+*Pia*+*Pid2*+*Pid3*+*Pita*	MR
23P-52	Guangdong	* Pia*+*Pid3*+*pikh*	R
23P-53	Guangdong	* Pia*+*Pid3*+*pikh*	R
23P-54	Guangdong	* Pi2*+*Pi5*+*Pid2*+*Pid3*	MR
23P-55	Guangdong	* Pi2*+*Pi5*+*Pia*+*Pita*	R
23P-56	Guangdong	* Pia*+*Pid2*+*Pid3*+*Pita*	MR
23P-57	Guangdong	* Pi2*+*Pi5*+*Pita*	R
23P-58	Guangdong	* Pi5*+*Pia*+*Pid2*+*Pid3*	MR
23P-59	Guangdong	* Pi5*+*Pid2*+*Pid3*	MR
23P-60	Guangdong	* Pi5*+*Pid2*+*Pid3*+*Pita*	MR
23P-61	Guangdong	* Pi2*+*Pi5*+*Pid2*+*Pid3*	R
23P-62	Guangdong	* Pi5*+*Pid2*+*Pid3*+*Pita*	MR
23P-63	Guangdong	* Pia*+*Pid2*+*Pid3*+*Pita*	MR
23P-64	Guangdong	* Pi2*+*Pita*	MR
23P-65	Guangdong	* Pi5*+*Pid2*+*Pid3*	MS
23P-66	Guangdong	* Pi5*+*Pid2*+*Pid3*+*Pita*	MS
23P-67	Guangdong	* Pi5*+*Pia*+*Pid2*+*Pid3*	S
23P-68	Guangdong	* Pid2*+*Pid3*+*Pita*	MS
23P-69	Guangdong	* Pi5*+*Pia*+*Pid2*+*Pid3*+*Pita*	MR
23P-70	Guangdong	* Pi2*+*Pi5*+*Pita*	MR
23P-71	Guangdong	* Pi2*+*Pi5*+*Pia*+*Pid2*+*Pid3*+*Pita*	MR
23P-72	Guangdong	* Pi2*+*Pi5*+*Pia*+*Pid2*+*Pid3*+*Pita*	R
23P-73	Guangdong	* Pi2*+*Pi5*	R
23P-74	Guangdong	* Pid2*+*Pid3*+*Pita*	R
23P-75	Guangdong	* Pi2*	MS
23P-76	Guangdong	* Pi5*+*Pid2*+*Pid3*+*Pita*	MS
23P-77	Guangdong	* Pi2*+*Pi5*+*Pid2*+*Pid3*+*Pita*	MS
23P-78	Guangdong	* Pi2*+*Pi5*+*Pita*	R
23P-79	Guangdong	* Pi5*+*Pid3*	MS
23P-80	Guangdong	* Pi2*+*Pi5*+*Pia*	MR
23P-81	Guangdong	* Pi5*+*Pia*+*Pid2*+*Pid3*+*Pita*	S
23P-82	Guangdong	* Pia*+*Pid2*+*Pid3*+*Pita*	S
23P-83	Guangdong	* Pi2*+*Pi5*+*Pia*+*Pita*	MS
23P-84	Guangdong	* Pi5*+*Pia*+*Pid2*+*Pid3*+*Pita*	MS
23P-85	Guangdong	* Pi5*+*Pia*+*Pid2*+*Pid3*	S
23P-86	Guangdong	* Pi5*+*Pia*+*Pid2*+*Pid3*	S
23P-87	Guangdong	* Pi5*+*Pid3*	MS
23P-88	Guangdong	* Pi2*+*Pita*	MR
23P-89	Guangdong	* Pi2*+*Pi5*	MR
23P-90	Guangdong	* Pi2*+*Pita*	MR
23P-91	Guangxi	* Pid2*+*Pid3*+*Pita*	MR

S: Susceptibility; MS: Moderate Susceptibility; MR: Moderate Resistance; R: Resistance.

**Table 2 plants-13-02134-t002:** Number of blast resistance genes and resistance evaluation in backbone varieties of *indica* rice.

The Number of Resistance Genes	The Number of Varieties	Resistance Frequency (%)	Resistance Grade of Varieties			
			R	MR	MS	S
1	5	40	0	2	3	0
2	17	58.82	2	8	7	0
3	32	62.5	10	10	12	0
4	21	66.68	3	11	3	4
5	12	75	3	6	1	1
6	4	100	1	3	0	0

**Table 3 plants-13-02134-t003:** Distribution of blast resistance genes combinations and accordingly resistance levels in backbone varieties of *indica* rice.

Gene Combination	Number of Varieties	Proportion (%)	Rice Blast Resistance Rate (%)
* Pia*+*Pid2*+*Pid3*	7	7.7	42.9
* Pia*+*Pid3*	1	1.1	0
* Pid3*+*Pita*	1	1.1	100
* Pia*+*Pid2*+*Pid3*+*Pigm*	1	1.1	100
* Pia*+*Pid2*+*Pid3*+*Pigm*+*pikh*	1	1.1	100
* Pia*+*Pid3*+*pikh*	3	3.3	66.7
* Pi1*+*Pi5*	1	1.1	0
* Pi2*+*Pid2*+*Pid3*	2	2.2	100
* Pi2*+*Pi5*+*Pita*	6	6.6	83.3
* Pi5*+*Pia*+*Pid2*+*Pid3*+*Pita*	9	9.9	77.8
* Pia*	2	2.2	50
* Pi5*+*Pi9*+*Pid2*+*Pid3*	1	1.1	100
* Pi5*+*Pi9*+*Pia*+*Pid2*+*Pid3*	1	1.1	100
* Pi5*	1	1.1	100
* Pi2*+*Pi5*	5	5.5	80
* Pid2*+*Pid3*	1	1.1	0
* Pi2*+*Pi5*+*Pia*+*Pid2*+*Pid3*+*Pita*	4	4.4	100
* Pia*+*Pigm*	1	1.1	100
* Pi5*+*Pia*	1	1.1	100
* Pi5*+*Pid2*+*Pid3*+*Pita*	5	5.5	60
* Pid2*+*Pid3*+*Pita*	6	6.6	50
* Pi5*+*Pid2*+*Pid3*	5	5.5	40
* Pi5*+*Pid3*	3	3.3	0
* Pi5*+*Pia*+*Pid2*+*Pid3*	6	6.6	50
* Pi2*+*Pi5*+*Pia*+*Pita*	3	3.3	66.7
* Pid3*	1	1.1	0
* Pi2*+*Pi5*+*Pia*	3	3.3	100
* Pi2*+*Pi5*+*Pid2*+*Pid3*	2	2.2	100
* Pia*+*Pid2*+*Pid3*+*Pita*	3	3.3	33.3
* Pi2*+*Pita*	3	3.3	100
* Pi2*	1	1.1	0
* Pi2*+*Pi5*+*Pid2*+*Pid3*+*Pita*	1	1.1	0

## Data Availability

The original contributions presented in the study are included in the article/Appendix A, further inquiries can be directed to the corresponding author.

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
