# Peer review of "Detection and Evaluation of Blast Resistance Genes in Backbone Indica Rice Varieties from South China"

_plants, 2024, doi:10.3390/plants13152134_

Round 1

Reviewer 1 Report (Previous Reviewer 2)

Comments and Suggestions for Authors

Dear Authors,

The changes you have made to the manuscript are sufficient. The clarity and scientific quality of the manuscript have been greatly improved.

Author Response

Comments 1: The changes you have made to the manuscript are sufficient. The clarity and scientific quality of the manuscript have been greatly improved.

Response 1:Thank you for your positive feedback on our manuscript. We are pleased to hear that the revisions have improved the clarity and scientific quality of our work.

Reviewer 2 Report (New Reviewer)

Comments and Suggestions for Authors

Dear all, the subject of the article is very important given the relevance of rice cultivation to the world. Manipulating the genes that confer resistance to rice blast is the most effective and ecologically sustainable means of controlling the disease.

My considerations are highlighted in the text. However, the way the document is displayed with previous markings and corrections as it is left it confusing and not very practical to act on.

Without further ado, I recommend the publication.

Author Response

Comment 1: The subject of the article is very important given the relevance of rice cultivation to the world. Manipulating the genes that confer resistance to rice blast is the most effective and ecologically sustainable means of controlling the disease.
Response 1: Thank you for recognizing the importance of our research on rice blast resistance. We agree that gene manipulation offers a promising and sustainable approach to disease control.
Comment 2: My considerations are highlighted in the text. However, the way the document is displayed with previous markings and corrections as it is left it confusing and not very practical to act on.
Response 2: We apologize for the confusion caused by the previous markings and corrections. We have revised the document to remove all previous annotations and provided a clean version for easier review. Please find the updated manuscript attached.
Comment 3: Without further ado, I recommend the publication.
Response 3: Thank you for your recommendation for publication. We are grateful for your positive feedback and support for our work.

This manuscript is a resubmission of an earlier submission. The following is a list of the peer review reports and author responses from that submission.

Round 1

Reviewer 1 Report

Comments and Suggestions for Authors

1.      In the manuscript, the author investigated the number of disease resistance genes and resistance responses using different rice genotypes. The results can provide information for gene pyramiding in rice breeding. However, in this study, the materials with more disease resistance genes do not seem to have better disease resistance, as 23P-81 and 23P-84, or the rice lines have the same genotypes, such as 23P-36 and 23P-41, but have different disease resistance performances. In addition, some materials are limited by the small number of samples, such as Pigm and pikh genes, and cannot be evaluated more objectively. These results should be studied and discussed in more depth.

2.      In the results, the authors mention that ‘…highly susceptible (HS) at 0, 20, 40, 27, 16, and 0 respectively’; I think the authors should provide a more detailed description of the data.

3.      In the results, P3, line 102, the authors mention 60 parents exhibiting resistance, but in Table 1, only 59 parents have blast resistance. Please check again.

4.      In Table 1, lines 23P-36 and 23P-41 have the same resistance genes, but the reaction differs. How to explain this phenomenon?

5.      In Figure 4, the authors should provide clear data information, such as Pi1, Pi9, Pigm, and pikh.

6.      It is suggested that authors should provide the functional markers information in the manuscript.

Comments on the Quality of English Language

None

Reviewer 2 Report

Comments and Suggestions for Authors

Dear Authors,

I have only a few comments. See the attached file below.
